# Monkeypox Clinical Features and Differential Diagnosis: First Case in Campania Region

**DOI:** 10.3390/pathogens11080869

**Published:** 2022-08-01

**Authors:** Novella Carannante, Claudia Tiberio, Raffaele Bellopede, Michela Liguori, Filomena Di Martino, Nicola Maturo, Raffaele Di Sarno, Sabrina Scarica, Giovanna Fusco, Lorena Cardillo, Claudio de Martinis, Luigi Atripaldi, Alessandro Perrella

**Affiliations:** 1Emergency Room Infection Disease Cotugno Hospital AORN dei Colli, 80131 Naples, Italy; novella.carannante@ospedalideicolli.it (N.C.); raffaele.bellopede@ospedalideicolli.it (R.B.); michela.liguori@ospedalideicolli.it (M.L.); filomena.dimartino@ospedalideicolli.it (F.D.M.); nicola.maturo@ospedalideicolli.it (N.M.); raffaele.disarno@ospedalideicolli.it (R.D.S.); sabrina.scarica@ospedalideicolli.it (S.S.); 2Virology Cotugno Hospital AORN dei Colli, 80131 Naples, Italy; claudia.tiberio@ospedalideicolli.it; 3Istituto Zooprofilattico Sperimentale del Mezzogiorno via Salute, 2, 80055 Portici, Italy; giovanna.fusco@izsmportici.it (G.F.); lorena.cardillo@izsmportici.it (L.C.); claudio.demartinis@izsmportici.it (C.d.M.); 4Pathology Central Laboratory Hospital AORN dei Colli, 80131 Naple, Italy; luigi.atripaldi@ospedalideicolli.it; 5First Division Emergency Infectious Disease and High Contagiousness Cotugno Hospital AORN dei Colli, 80131 Naples, Italy

**Keywords:** monkeypox, Varicella, Chickenpox, pandemic, infection control

## Abstract

As of 15 June, there have been, globally, a total of 2103 laboratory-confirmed cases and one probable case of Monkeypox, including one death. We report two cases of vesicular infectious diseases, one of those is the first case of Monkeypox in the Campania Region. The report, therefore, highlights a recrudescent infection disease that could represent a challenge in differential diagnosis with other vesicular infectious diseases such as Varicella Zoster Virus, during a pandemic season that does not seem to end. Indeed, varicella should be carefullu considered in differential diagnosis according to its vesicular or pustular rash to have a prompt diagnosis and public health response in case of monkeypox infection.

## 1. Introduction

During the second week of June 2022 at 12:00 AM, a 40-year-old man was admitted to the Infectious Disease Emergency Room (IDER) at Cotugno Hospital in Naples. He reported clinical features for the past three days and fever for one day (with the highest peak at 38.2 °C) with the onset of asynchronous and mild vesicular rash mainly on the trunk (Figure 1), and also genital and perineal pain. On the day of IDER admission, he only suffered intense headaches and myalgia without further fever episodes but was responsive to paracetamol. Upon clinical examination, we found several pustules, about 2–4 mm each, on an erythematous base with central umbilication interesting the face, neck, abdomen and hands. A few 2–3 mm round painful erosions were noted in the perineal area while an intact pustule was observed in the lower abdominal region (Figure 1b). We did not observe cervical lymphadenopathy, pneumonia signs or the involvement of abdominal organs. According to the vesicle onset, the patient reported the first one localized on the scalp about 5 days before and being in the crustification phase at the moment of clinical examination. He did not report any significant risk factor; he had not traveled abroad to countries at risk or had any contact with possible cases or subjects being confirmed as monkeypox-infected. On the contrary, he reported a relative absence of social contact in the previous 14 days from the onset of fever. He only reported local travel to work in the surrounding area of Naples. Despite the patient not presenting any risk, in consideration of the current monkeypox outbreak [1], we decided to manage him according to our internal protocol for differential diagnosis for Monkeypox or Varicella Zoster Virus (VZV) and therefore he underwent our protocol for Emerging Infectious Disease (Figure 2). The laboratory tests did not show any increase in inflammation markers such as PCR, IL-6, PCT and WBC, we only found a mild increase in the interleukin 2 receptor IL2R. No other significant blood parameters alterations were found; neither positive IgM or IgG for Chickenpox (Varicella zoster Virus—VZV) (Table 1). A molecular test for Monkeypox was found to be positive on vesicles and on a nasal swab specimen. The test was based on viral DNA extraction with a Qiamp Viral RNA mini kit (Qiagen, Italy Branch, Milan, Italy) and two real-time PCRs were used to assess the presence of MPXV DNA. A Real-Star Orthopoxvirus PCR Kit (Altona Diagnostics GmbH, Hamburg, Germany) was used as the screening PCR. This method recognizes a region common to all Orthopoxviruses without distinction of species. The second PCR (G2R_G assay) is based on previous evidence [2]. Notably, a further real-time PCR assay was carried out according to the protocol of Li and colleagues (2010) that allows for the detection of generic MPXV DNA and further differentiation between West African and Congo Basin strains [2]. The reactions were carried out in a 25 µL final volume, containing KAPA PROBE FORCE qPCR Master Mix Universal 1× (Kapa Biosystems Pty, Cape Town, South Africa), 0.4 µM for each primer set and 0.2 µM for probes (FAM-labeled). An internal control (0.4 µM final concentration) was added to exclude any possible inhibition using Beta Actin Mix (VIC-labeled). The thermal profile included enzyme inactivation/template denaturation at 98 °C for 3 min, followed by 45 cycles of denaturation and annealing/extension at 95 °C for 5 s and 60 °C for 20 s, respectively, for MPXV and the Congo Basin strain, while the annealing/extension phase was 62 °C for 20 s for the West African strain. The amplifications were performed on a QuantStudio 5 real-time PCR system (Applied Biosystems, Foster City, CA, USA) thermal cycler. Positive results were obtained for MPXV and for West African lineage, showing threshold cycle (Ct) values of 23 for both the real-time PCR assays (Figure 3). The sample was then submitted to an end-point PCR modifying a protocol already described [3], thus using the MPXV-ext_FOR forward primer of the protocol of Dumont and colleagues, while we designed the reverse primer (MPXV_REV_2: 5′-ATCCATGTATTGCGCCAAATA-3′) giving rise to a 571 bp amplicon. The reaction was carried out in a 25 µL volume, including Kapa2G Robust HotStart Ready Mix 1× (Kapa Biosystems), along with 0.2 µM final concentration for each primer and 5 µL template. The amplification was performed on Mastercycler Nexus X2 thermal cycler (Eppendorf) applying the following thermal cycle, 95 °C for 3 min for activation and 40 cycles at 94 °C for 30 s for denaturation, 57 °C for 30 s for annealing and 72 °C for 30 s for extension, followed by the last extension cycle at 72 °C for 10 min. Next, 1 µL of the amplification product was used for the capillary electrophoresis (Tapestation 2200, Agilent Technologies, Santa Clara, CA, USA) with D1000 screen tape and reagents, followed by Sanger sequencing, carried out with a Big Dye Terminator Cycle Sequencing Kit v.1.1 (Applied Biosystems, Warrington, UK). Finally, the reaction was applied to a 3500 Genetic Analyzer capillary electrophoresis system (Applied Biosystems). The forward and reverse sequences were assembled using the Geneious R9 software package (Biomatter, Auckland, New Zealand) and compared to analogous sequences in the BLAST genetic database (http://www.ncbi.nlm.nih.gov/Blast.cgi accessed on 21 July 2022 A phylogenetic analysis was carried out using Mega X software [4]. The evolutionary history was inferred by using the Maximum-Likelihood method and the Jukes–Cantor model (5). The bootstrap consensus tree inferred from 1000 replicates [5] is taken to represent the evolutionary history of the taxa analyzed [6]. Branches corresponding to partitions reproduced in less than 50% bootstrap replicates are collapsed. The percentage of replicate trees in which the associated taxa clustered together in the bootstrap test (1000 replicates) are shown next to the branches [6]. Initial tree(s) for the heuristic search were obtained by applying the neighbor-joining method to a matrix of pairwise distances estimated using the Maximum Composite Likelihood (MCL) approach. A discrete Gamma distribution was used to model evolutionary rate differences among sites (4 categories (+G, parameter = 0.0500)). The rate variation model allowed for some sites to be evolutionarily invariable ([+I], 15.11% sites). This analysis involved 18 nucleotide sequences. Codon positions included were first, second, third and non-coding. The sequence was compared to other orthopoxviruses, along with Central African and West African reference genomes. Results confirmed that our sample belonged to the West African clade (Figure 4). The reactions were carried out in a 25 µL final volume, containing KAPA PROBE FORCE qPCR Master Mix Universal 1X (Kapa Biosystems Pty, Cape Town, South Africa), 0.4 µM for each primer set and 0.2 µM for probes (FAM-labeled) (2) (Figure 3). According to the results, based on the Italian Ministry of Health law and based on clinical presentation, the patient opted for home isolation with a clinical follow-up by the Local Health Unit after a fluid in vein therapy with paracetamol for the headache. Following the first Monkeypox case in the Campania Region, the next day we had a new admission in our IDER of a 75-year-old man with a recent onset of vesicles starting from the scalp and spreading to the trunk, arms, palm of the hands and legs (Figure 5). He did not report fever nor had other clinical symptoms on the previous day of the vesicles occurrence. As with the first case, he had not returned from foreign travel, nor had he had contact with people suffering from signs or symptoms of infection. At clinical examination, he did not show any increase in lymph nodes. A chest clinical examination showed the presence of crackles in the pulmonary area with an absence of abdominal organ involvement upon further clinical examination. According to the age and clinical features, a lung CT scan was performed but it did not show any significant signs of pneumonia. Following our internal protocol, like with the first patient, all laboratory tests were performed showing the absence of any significant inflammatory marker and being negative for Monkeypox but positive for (VZV) IgM (Table 1). After receiving a prescription for acyclovir the patient preferred home isolation with a local health unit follow-up.

## 2. Discussion

The outbreak of Monkeypox seems to continue to primarily affect men who have sex with men (MSM) and who have reported recent sex with new or multiple partners, suggesting that close contact through sexual intercourse could be a cause of spread [1,7,8]. As of 15 June, there have been a total of 2103 laboratory-confirmed cases and one probable case of Monkeypox, including one death, reported to the WHO, with the majority of cases occurring since May 2022 [8]. Human Monkeypox is a zoonotic Orthopoxvirus with a double-stranded DNA virus of the *Orthopoxvirus* genus of the *Poxviridae* family. Two genetic MPXV clades were characterized: West African and Central African. Outside of Africa, the first cases of monkeypox were reported in 2003 in the United States (US). Currently, European countries are experiencing an outbreak of Monkeypox cases in their territories, while infections caused by a persistent SARS-CoV-2 pandemic are still ongoing. In Italy, at the time of our monkeypox evaluation, we registered 48 cases, most of them without clinically significant manifestation [8,9]. Therefore, the eCDC and the WHO are underlining the importance of follow-up on the occurrence of these cases when diagnosed [1,8]. Although the public health experience addressing Monkeypox in Italy and in the Campania Region is limited, this case illustrates mainly two fundamental aspects of the effectiveness of infection control: Rapid identification of the pathogen (differential diagnosis);Early isolation protocol of the index patient with significant support from the laboratory for a prompt differential diagnosis.

Indeed, clinical features of Monkeypox could be sometimes confounding, particularly alongside the Varicella Zoster Virus (VZV). It has recently been underlined that diagnosis of Monkeypox could be a challenge, according to possible confounding factors propending for a VZV diagnosis too [7]. Indeed it is well known that lesions on the palms of the hands and soles of the feet are often noted in MPX patients; although this feature is not recognized as a significant VZV clinical feature, it can be present sometimes. In our case report, including the first case of Monkeypox infection in the Campania Region, both patients presented lesions on the palm of the hands with irregular borders that were very similar in terms of manifestations and therefore, clinical diagnosis was a challenge. 

Particularly the second patient, according to laboratory tests, would seem to have experienced a primary VZV infection, which would also seem an unlikely clinical event considering the age. 

Consequently, differential diagnosis with Varicella only based on clinical presentation would have been difficult and would not be enough in a fast-track infection control strategy. 

Indeed, early and quick laboratory diagnostics seem to be mandatory to perform a more exhaustive evaluation, particularly in some epidemiological settings or cohorts of patients (10). 

According to our experience and what the WHO and eCDC are suggesting in their report [1,2], it is our opinion that a fast VZV marker assay should always be performed in ER to quickly address the patients to the best diagnostic therapeutic approach. In fact, in Italy and in the Campania Region, despite having had about 88.5% of VZV vaccine coverage since 2017, we should consider a possible failing vaccine coverage as well as non-vaccinated people as confounding factors [10,11]. Another significant consideration, despite being based on only one case, is that given the possible human-to-human transmission even in the absence of travel in endemic areas or close contact for sexual intercourse, it would suggest that currently, local Monkeypox diffusion could be underestimated and therefore risk factors could not be only related to what is being currently evaluated. Thus, it could indicate that international, local and regional support for increased surveillance and detection of Monkeypox cases are essential tools for understanding the continuously changing epidemiology of this resurging disease. The waning population immunity associated with the discontinuation of the smallpox vaccination and the emergency COVID-19 pandemic would have established a landscape for the resurgence of Monkeypox that could really represent a significant issue for public health [12]. In conclusion, we reported the first case of Monkeypox in Campania that highlights a recrudescent infection disease that could represent a challenge in the differential diagnosis. Particularly, VZV should be considered different from that of a vesicular or pustular genital rash and requires prompt diagnosis and public health response. Certainly, when Monkeypox is suspected while waiting for other laboratory tests, patients should be treated with an infection control-based strategy, isolated and once diagnosed, close contacts should also be traced. 

## Figures and Tables

**Figure 1 pathogens-11-00869-f001:**
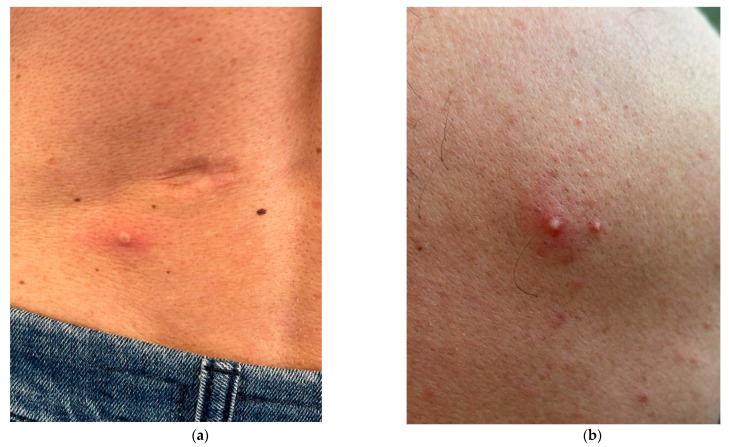
(**a**) A 2–4 mm pustule with an erythematous base with central umbilication interesting lower abdomen; (**b**) Two close pustules with an erythematous base on the trunk.

**Figure 2 pathogens-11-00869-f002:**
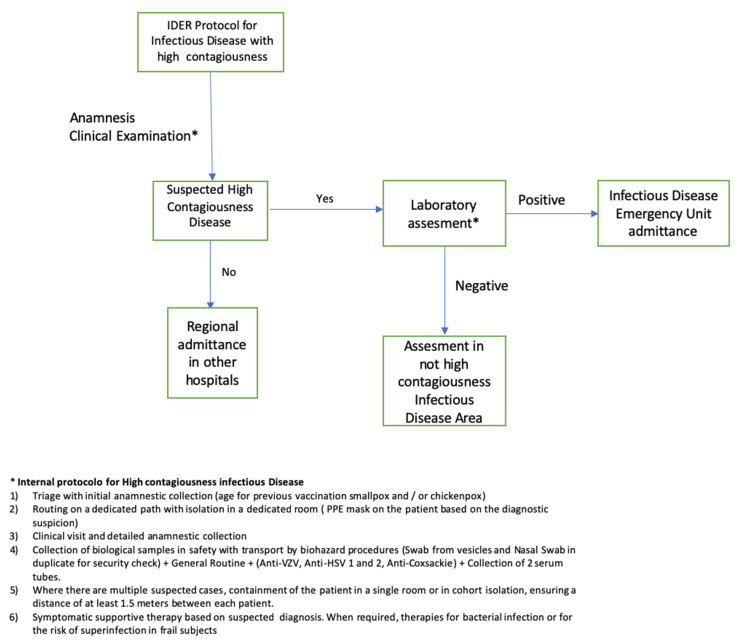
Infection Control alghoritm for patients admitted in IDER (Infectious Disease Emergency Room) at AORN Ospedali dei Colli.

**Figure 3 pathogens-11-00869-f003:**
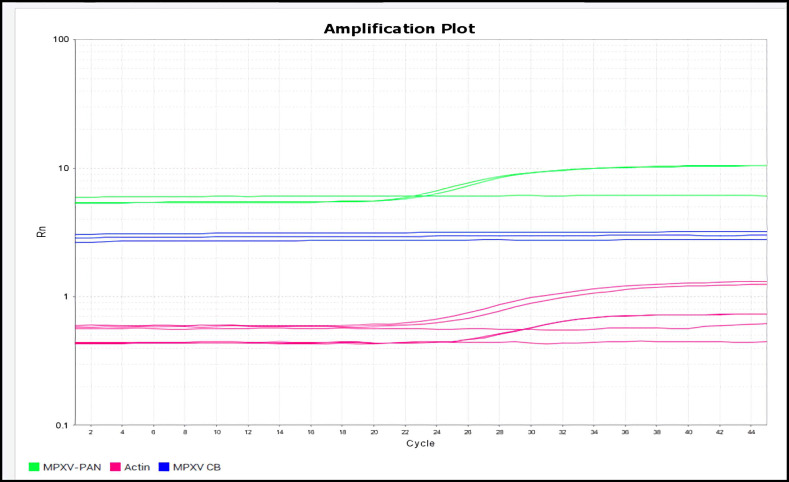
A Real-time PCR amplification plot for generic Monkeypox virus and Congo Basin strain; b Real-time PCR amplification plot for West African strain. (**a**) shows positive results for generic Monkeypox virus (green) and negative results for Congo Basin strain (blue). The amplification of beta actin internal control (purple) is visible demonstrating no PCR inhibition. The reaction was performed in duplicate. (**b**) shows a real-time PCR linear plot showing the positive results for West African strain (green) and beta actin internal control (blue) demonstrating no PCR inhibition. All reactions were performed in duplicate.

**Figure 4 pathogens-11-00869-f004:**
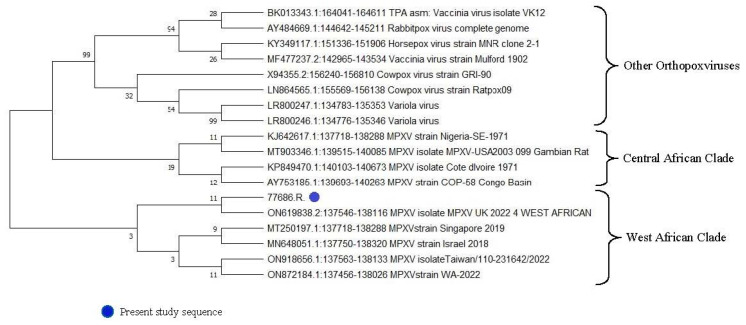
Phylogenetic analysis for Monkeypox virus.

**Figure 5 pathogens-11-00869-f005:**
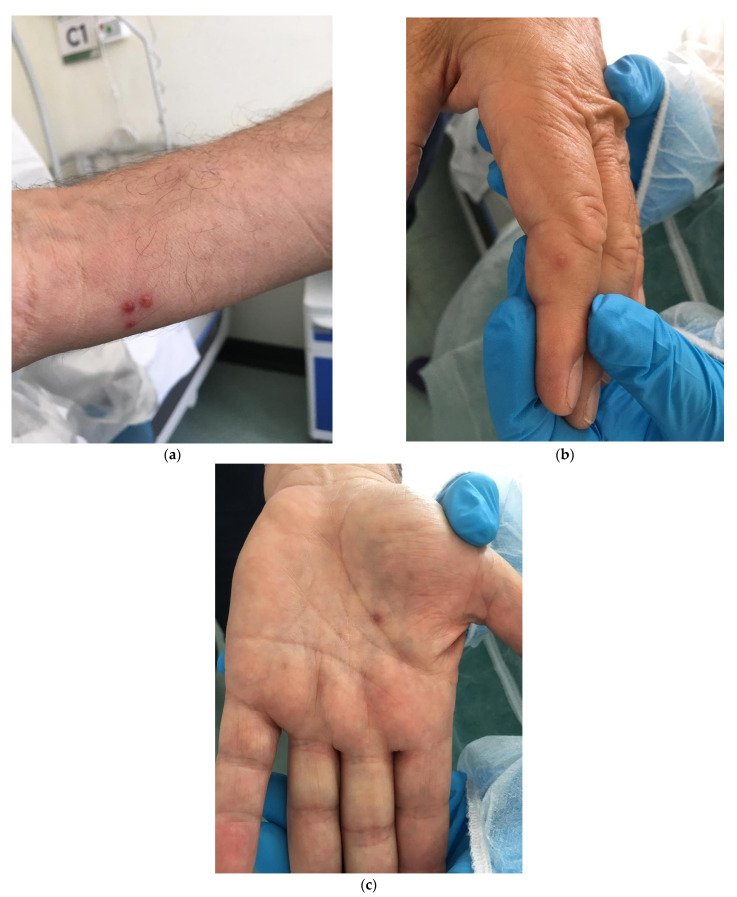
(**a**): A 2–3 mm pustules with small erythematous base; (**b**): A vesicle on the finger of right hand with central umbilication; (**c**): A vesicle on the palm of the hand. Figure 5 shows vesicles on patient being negative for Monkeypox but positive for Chickenpox—VZV.

**Table 1 pathogens-11-00869-t001:** Descriptive parameters of two CASES.

Characteristics	Patient 1	Patient 2
Sex	Male	Male
Age (years)	40 yrs	75 yrs
Previous STIs	None	None
Recent sexual exposure	No	No
Systemic symptoms	Fever, headache	none
Days from systemic symptoms to appearance of lesion	1	NA
Localization of skin lesions	Genital, thorax, scalp, trunk, abdomen, perineal area	Back, legs,
		foot sole, hand, scalp, trunk, abdomen
Evolution of lesions	Asynchronous	Asynchronous
Laboratory Findings		
WBC (cell/mmc)	6090	7050
Monocytes	480	670
Lymphocytes	1770	3080
AST/ALT (<40 UI/mL)	20/54	34/48
LDH (250 UI/mL)	235	227
INR	1,31	1,24
CRP	0,2	0
IL2R(223–710 IU/mL)	894,000	737,000
IL6 (0–5 pg/mL)	4,5	n.d *
VZV IgM/IGG	−/+	+/−
HSV1/HSV2-DNA	−/−	−/−
SARS-CoV-2 IgG	+	+
SARS-CoV-2 RNA	−	−
MonkeyPox DNA	+	−

Table shows major clinical and laboratory features of both patients. * means “not determined” result of evaluated laboratory parameter.

## Data Availability

No data are available since it is a Case report.

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
