# Peer review of "Monkeypox Clinical Features and Differential Diagnosis: First Case in Campania Region"

_pathogens, 2022, doi:10.3390/pathogens11080869_

Round 1

Reviewer 1 Report

The case report by Novella Carannante et al described confirmed cases of Monkeypox in Campania and advocated VZV in differential diagnosis with another vesicular infectious disease. Fast and confirmatory diagnosis of emerging pathogens is the need of the prevailing and current situations.  However, there are a few important points that need to be addressed.

1)    A graphical representation of the targeted (Monkeypox) transcript normalized to beta actin internal control in the Real-time PCR will be more illustrative than that in Figure 3.

2)    Owing to the current scenario, a phylogenetic tree of the two conformed positive samples of monkeypox, characterized against the available database should be provided.

Author Response

Dear Referee

Thank you very much for your precious observations

according to your suggestion we have added:

  1. A graphical representation using the normalized Rn plot was replaced within the manuscript.
  2. The phylogenetic analysis was added along with the phylogenetic tree. As soon as we will obtain the accession n. from GeneBank database, we will insert it in the manuscript.

Thank you again

Reviewer 2 Report

The present manuscript lacks novelty and is not apropriate for pathogens. It contains an incomplete case description of monkeypox, and it repeats general statements of differential diagnostics. As the subject needs attention in general practice,  I would suggest to submit the manuscript to regional (Italian) journal of dermatology or general medicine.

Author Response

Dear Referee

We are sorry about your review.

Currently, 21/07/2022, there are about 47 case reports on MonkeyPox some of those are based  on first case in some regions. The case report descriptions in litterature could be helpful particularly for this kind of infectious disease that presents some confounding features like VZV. We described at the best of clinical presentation that was confounding. Therefore we would like to publish this case report that could be, as for other recent papers in litterature, useful for clinicians and physician to better intercept patients with monkeypox admitted to ER.

Reviewer 3 Report

I have the following comments to further strengthen the manuscript.

Abstract: it should be improved

Section "Introduction": does it begin with "On June 2022 at ...."?

Introduction: what about local treatment of the skin lesions? Could you describe it for both patients?

You use the acronymous ER, but it seems that the right one is IDER.

In Figure 2 the legend * refers to both steps "Anamnesis Clinical Examination" and "Laboratory Assessment", right?

Maybe better explanatory titles and captions of the figures should be given. 

Discussion: what about the origin of MPXV infection? Could you write some hypotheses?

Broadly, the manuscript summarizes an interesting case report on the diagnosis of a Monkeypox infection in men in Campania Region.

Author Response

Dear Reviewer, Thano you very much for your effort and assistance

We strongly appreciated your suggestions and criticisms. Please consider that all modification are in yellow. Further x we managed the revision as follows:

Abstract: it should be improved. We impoved abstract

Section "Introduction": does it begin with "On June 2022 at ...."? We changed the Incipit

Introduction: what about local treatment of the skin lesions? Could you describe it for both patients? We detailed about that

You use the acronymous ER, but it seems that the right one is IDER. We corrected the typewriting error

In Figure 2 the legend * refers to both steps "Anamnesis Clinical Examination" and "Laboratory Assessment", right? Yes

Maybe better explanatory titles and captions of the figures should be given.  WE tried to improve title

Discussion: what about the origin of MPXV infection? Could you write some hypotheses? WE wrote some possible explaination, particularly we inderlines the possible understimation of  current monkeypox cases around european countries